# Use of administrative data for evaluating trends in medically-attended Lyme disease, Manitoba, Canada, 2010–2021

Maria Major[1*], Susan Horton[2], Natalie Nightingale[3], Kate Halsby[4], Frederick J. Angulo[5], James Stark[6], Holly Yu[7], Mark Loeb[8], Irene Wang[3], Saranya Nair[3], Calum S. Neish[3], Doneal Thomas[3], Chloe McDonald[1], Samuel Torres-Florez[1,9], Ana G. Grajales[1], Sarah J. Willis[6]

**1** Pfizer, Kirkland, Quebec, Canada, **2** University of Waterloo, Waterloo, Ontario, Canada, **3** IQVIA, Mississauga, Ontario, Canada, **4** Pfizer, Tadworth, United Kingdom, **5** Pfizer, New York, New York, United States of America, **6** Pfizer, Cambridge, Massachusetts, United States of America, **7** Pfizer, Collegeville, Pennsylvania, United States of America, **8** McMaster University, Hamilton, Ontario, Canada, **9** Department of Bioengineering, McGill University, Montreal, Quebec, Canada

\* Maria.major@pfizer.com

## Abstract

While Lyme disease (LD) is the most common tick-borne disease reported to public health surveillance in North America and is increasingly recognized as a public health threat in Canada, it's incidence may be underreported. Our study aims to estimate medically-attended LD incidence in Manitoba, Canada, using administrative health data. We identified medically-attended LD cases in Manitoba from 2010 − 2021 in a claims database (Manitoba Population Research Data Repository), which contains the health records of >95% of the residents of Manitoba, using diagnostic codes, antibiotic dispensations, and laboratory results. The incidence of claims-based LD cases ranged from 8.4 to 28.5 per 100,000 population per year, 5.1 to 11.0 times higher than the incidence of surveillance-reported LD cases. The incidence of claims-based LD cases was particularly higher than the incidence of surveillance-reported LD cases in females and health regions with a low surveillance-reported incidence. Our study suggests that medically attended LD is more common than reported in surveillance. Further study is required to identify barriers to reporting. Interventions are needed to reduce the substantial burden of LD in Manitoba.

## Introduction

Lyme disease (LD) is the most common tick-borne disease reported in Canada. *Borrelia burgdorferi* sensu stricto *(B. burgdorferi)*, the causative agent of LD, is transmitted by *Ixodes scapularis* (*I. scapularis*) in central and eastern Canada, and by *Ixodes pacificus* (*I. pacificus*) in British Columbia [1,2]. LD is a multi-system disease with

which permits unrestricted use, distribution, and reproduction in any medium, provided the original author and source are credited.

**Data availability statement:** All relevant data are within the paper and its Supporting information files.

**Funding:** This study was supported by Valneva and Pfizer in the form of a grant awarded to ML, fees to the Manitoba Population Research Data Repository and IQVIA, and Pfizer in the form of a salary for MM, KH, FJA, JS, HY, CM, AG, and SJW. The specific roles of this author are articulated in the 'author contributions' section.

**Competing interests:** The authors have read the journal's policy and have the following competing interests: MM, KH, FJA, JS, HY, CM, AG, and SJW are employees of Pfizer. This does not alter our adherence to PLOS ONE policies on sharing data and materials. There are no patents or marketed products associated with this research to declare, however Valneva and Pfizer are co-developing a vaccine for the prevention of Lyme disease.

clinical manifestations categorized into 3 stages, early localized LD, early disseminated LD, and late disseminated LD [3,4]. Early localized LD usually presents as a bullseye rash, known as *erythema migrans* (EM), and non-specific symptoms, such as malaise [3,4]. Early disseminated LD presentations include multiple EM lesions, neurologic manifestations, aseptic meningitis, cranial neuropathy (Bell's palsy), and Lyme carditis [3,4]. Late disseminated LD presentations, which may occur months to years after infection, include Lyme arthritis, Baker's cyst, meningitis, and subacute mild encephalopathy [4].

LD became a notifiable disease in Canada in 2009. Physicians and laboratories reported LD cases to provinces and territories, and provinces and territories report cases to the Public Health Agency of Canada (PHAC) through the Canadian Notifiable Disease Surveillance System (CNDSS) [2]. The number of surveillance-reported LD cases have risen in Canada from 144 in 2009–3,147 in 2021, with 95.6% of reported cases in the eastern provinces of Ontario, Québec, and Nova Scotia [5]. Canadian provinces and territories also conduct tick surveillance; however, methods and geographic scope are inconsistent and therefore do not provide a comprehensive assessment of the tick-bite risk [6,7]. Manitoba has the fifth highest incidence of surveillance-reported LD cases in Canada, with 3.0 LD cases per 100,000 population in 2021 [5].

There has been limited research conducted to assess the magnitude of underreporting medically-attended LD cases to public health reporting [8,9]. Claims databases have been frequently used in the United States (US) to estimate the incidence of medically-attended LD cases. By using Lyme-specific billing codes and antibiotic prescriptions, Kugeler et al (2021) estimated that there were ≈476,000 patients in the United States diagnosed with LD annually during 2010–2018 [10]. Using similar methods, Schwartz et al (2021) estimated that the incidence of medically-attended LD cases was 6–8 times higher than the incidence of surveillance-reported LD cases [11]. A study by Cocoros et al. (2023) used clinicians to review the medical records of LD cases identified in a claims database and demonstrated that 93.8% (95% CI 88.1%–97.3%) of the claims LD cases met a clinical definition of LD [12]. A recent study by Rusk et al. (2025) used administrative data to estimate an incidence of medically attended LD of 10.2 per 100,000 population in Manitoba from 2009–2018 [13]. These studies demonstrate that claims databases may be used to estimate the incidence of medically-attended LD.

Due to universal access to the publicly funded healthcare system in Canada, almost all healthcare visits have the potential to be included in the claims databases of the healthcare system. However, each province independently governs healthcare, leading to variability in data content, data quality, and governance, which is usually in the form of an academic data custodian that restricts access to administrative health databases and may limit access for research purposes [14]. The aim of our study was to use previously validated methods to estimate the incidence of medically-attended LD in the Manitoba claims database, which is a comprehensive provincial health record registry, and to compare that incidence to the incidence of surveillance-reported LD cases [15].

## Methods

### Data sources

The data was sourced from Manitoba Population Research Data Repository (MPRDR), from August 2023 to January 2024. MPRDR is a provincial database that contains administrative health records for the Manitoba population; populations not covered include temporary visitors (visiting less than 6 months), and persons residing on military facilities or First Nations reserves [15]. In July 2024, there were 1.35 million individuals with active records in the MPRDR [15,16]. The MPRDR includes data related to medical visits at hospitals, physicians offices, emergency departments, specialists, homecare, diagnostic testing results and pharmaceutical prescriptions dispensed, linked through individual Personal Health Identification Numbers (PHIN) (S1 Table in S1 File) [15].

### Identification of LD cases from MPRDR

Medically-attended LD cases were identified in the MPRDR claims database in 2010−2021 using four case finding algorithms. The first algorithm, applied to persons seeking care at an emergency room, identified medically attended LD cases as individuals with an International Statistical Classification of Diseases and Related Health Problems, Tenth Revision, Canada (ICD-10-CA) code for LD who received a ≥ 7-day course of antibiotics dispensed within 30 days of the emergency room visit. The ICD-10-CA codes were extracted from the National Ambulatory Care Reporting System (NACRS) and drug dispensations were extracted from the Drug Program Information Network (DPIN). The second algorithm identified hospitalized patients with LD in the Discharge Abstract Database (DAD), which is a product of Canadian Institute of Health Information (CIHI), as individuals with at least one Lyme-specific ICD-10-CA code. We report emergency room LD cases and hospitalized LD cases together. The third algorithm, applied to persons attending primary care clinics, identified medically-attended LD cases as individuals with International Classification of Diseases, 9th Revision, Clinical Modification (ICD-9-CM) code for LD who received a ≥ 7-day course of antibiotics dispensed within 30 days of the primary care clinic visit. ICD-9-CM codes were extracted from the Medical Claims-Medical Services (MC-MS) database and antibiotics were extracted from the DPIN. The fourth algorithm identified individuals as LD cases if they had positive diagnostic serology results from the Cadham Provincial Laboratory (CPL), based upon positive two-tier test results using CDC criteria for interpretation of western blots [17]. Detailed descriptions of LD algorithm definitions and claim codes used are available in S2–S5 Tables in S1 File.

Identified LD cases were assigned a date of LD diagnosis as the earliest date they were identified as a case by any of the LD case-finding algorithms. Patients were assigned unique identifiers by MPRDR, which prevented the risk of double counting. If individuals met more than one algorithm definition during the selection period, they were indexed on the earliest day they met any of the algorithm definitions. If individuals met more than one algorithm definition on the same day, they were indexed on the algorithm per following order of precedence: emergency room or inpatient algorithm, primary care-algorithm, serology-based algorithm, to register the most severe outcome. Patients were excluded if key demographics (e.g., sex, age) were missing, or age was ≥ 105 years.

### Demographic and clinical variables

The demographic and clinical variables of interest included age, sex, area of residence, neighborhood income quintile, seasonality, year of diagnosis, and clinical stage. The calculated age at index was presented as a categorical variable with the following categories: ≤ 10, 11–20, 21–30, 31–40, 41–50, 51–60, 61–70, 71–80, and ≥ 81 years. The area of residence refers to five regional health authorities (RHA) in Manitoba: Northern Regional Health Authority (NRHA), Interlake-Eastern Regional Health Authority (IERHA), Prairie Mountain Health (PMH), Winnipeg Regional Health Authority (WRHA), and Southern Health – Santé Sud (SHSS). The neighborhood income quintile was derived using the Statistics Canada Postal Code Conversion File. LD cases were classified into one of four clinical staging categories based on symptoms occurring

within 90 days before or after the index date: early localized stage, early disseminated stage, late disseminated stage, and undefined (S6–S9 Tables in S1 File) [18].

### Data privacy and ethics statement

The data used for this study were extracted by the MPRDR, under the stewardship of the University of Manitoba and Manitoba Centre for Health Policy (MCHP), in compliance with data management and data privacy policies, as per MPRDR internal protocols. All data outputs received were at an aggregate level, using pre-submitted table shells provided to MPRDR. All data were fully anonymized before we accessed them. The ethics committee waived the requirement for informed consent. To protect patient privacy and potential de-identifying of data, any data cell values with a count of less than 6 patients were suppressed, along with the second smallest group to prevent back-calculation. No individual-level data were transferred outside of MPRDR data servers. This study underwent ethics approval by the University of Manitoba, Health Research Ethics Board (#HS25991).

### Data analysis

LD incidence was defined as the number of claims-based LD cases per 100,000 population per year using mid-year census population estimates from Statistics Canada [19]. The incidence per 100,000 population per year of surveillance reported LD cases was calculated for 2010–2021. Where possible, analyses were stratified by age group, RHA, and sex. All calculation and analyses were conducted using Microsoft Excel, SAS version 9.4 or higher (SAS Institute Inc., Cary, NC, USA) or R, version 4.2.0 or later (The R Foundation for Statistical Computing, Vienna, Austria).

## Results

### Claims-based LD cases

A total of 2,976 medically-attended LD cases were identified in the claims database from 2010–2021, of which 2912 had complete demographic data; 1364 (46.9%) LD cases were female and 1548 (53.2%) male (Table 1). The median age of the medically-attended LD cases was 51.0 years, interquartile range (IQR) 29.5–64.0 years (Table 1). By income strata, 24.5% of medically-attended LD cases were in the highest income quintile (Table 1). The proportion of claims-based LD cases indexed by algorithm were as follows: primary care algorithm (80.1%), diagnostic serology (13.6%), and EDs or hospitalizations (6.3%) (Table 1). Clinical staging of LD cases could not be assessed, as between 90–100% of the claims-based LD cases were categorized as "undefined" (S10 Table in S1 File).

The average incidence of medically-attended LD cases was 18.4 cases per 100,000 population per year during 2010–2021, with an increase from 8.4 cases per 100,000 population in 2010 to 28.5 cases per 100,000 population in 2019 (Table 2). The average incidence of medically-attended LD during 2010–2021 was 17.0 cases per 100,000 population per year for females and 19.7 cases per 100,000 population per year for males (Table 2). Medically-attended LD cases were primarily diagnosed from May to August (Fig 1).

The medically-attended LD cases were distributed by RHA as follows: 1175 (40.6%) in WRHA, 912 (31.3%) in SSHS, 426 (14.6%) in IERHA, 375 (12.9%) in PMH, and 24 (0.8%) in NRHA (Table 1). In the 2 regions with complete reporting from 2010–2021, the average incidence of medically-attended LD was 13.4 per 100,000 population per year in WRHA and 38.4 per 100,000 population per year in SHSS (S12 Table in S1 File). The overall incidence of medically-attended LD cases from the claims database was 5.1 to 11.0 times higher than the incidence of surveillance-reported LD cases (Table 2).

## Discussion

Our results indicate that for each surveillance-reported LD case in Manitoba, there are between 5.1–11.0 additional medically-attended LD cases that are not reported. This was observed in each of the 5 public health regions in the Manitoba except NRHA which is the northernmost region in the province. In Manitoba, we observed higher underreporting of

**Table 1. Characteristics of Medically-Attended Lyme Disease (LD) Cases.**

| | Category | N (%) |
|---|---|---|
| | **Medically Attended LD Cases** | **2912 (100)** |
| Sex | **Females** | 1364 (46.8) |
| | **Males** | 1548 (53.2) |
| Age Group (years) | **≤ 9** | 309 (10.6) |
| | **10 - 19** | 209 (7.2) |
| | **20 - 29** | 210 (7.2) |
| | **30 - 39** | 304 (10.4) |
| | **40 - 49** | 367 (12.6) |
| | **50 - 59** | 524 (18.0) |
| | **60 - 69** | 538(18.5) |
| | **70 - 79** | 339(11.6) |
| | **80 - 89** | 106 (3.6) |
| | **≥90** | 6 (0.2) |
| | **Mean Age (Standard Deviation)** | 46.2 (22.9) |
| | **Median Age (1st Quartile, 3rd Quartile)** | 51.0 (29.5, 64.0) |
| | **Age Range (Minimum – Maximum)** | 1.0 - 96.0 |
| Regional Health Authorities | **Northern Regional Health Authority** | 24 (0.8) |
| | **Interlake-Eastern Regional Health Authority** | 426 (14.6) |
| | **Prairie Mountain Health** | 375 (12.9) |
| | **Winnipeg Regional Health Authority** | 1175 (40.4) |
| | **Southern Health – Santé Sud** | 912 (31.3) |
| Neighborhood income quintile | **1 (lowest)** | 431 (14.8) |
| | **2** | 487 (16.7) |
| | **3** | 630 (21.6) |
| | **4** | 644 (22.1) |
| | **5 (highest)** | 714 (24.5) |
| | **Unavailable** | 6 (0.2) |
| Algorithm | **Primary Care** | 2383 (80.1) |
| | **Serology** | 406 (13.6) |
| | **Hospitalized/Emergency Department** | 187 (6.3) |

LD cases in females, which was also observed in Schwartz et al [11]. This study provides valuable insights into under-ascertainment of medically attended LD in a province with a low incidence of surveillance-reported LD cases in Canada.

Our findings were consistent with results from other studies using similar methodology. Our study identified a higher number of claims-based LD cases from 2010–2018 in Manitoba than was reported in Rusk et al. (2025) [13]. This difference could partially be accounted for by our additional case-finding algorithm which identified LD cases using diagnostic serology results of patients whose medical records did not register a diagnostic Lyme code. This algorithm resulted in an additional 406 LD cases over our entire 11-year study period and 266 additional LD cases from 2010–2018, which were not captured by using diagnostic Lyme and treatment codes.

A US study by Schwartz et al. (2023), which also used administrative data to estimate the LD incidence, reported 6–8 times higher LD incidence than was reported to surveillance [11]. Schwartz et al. (2023) observed higher under-reporting of claims-based LD cases in states that had low surveillance-reported LD incidence, compared to states that reported higher LD incidence [11]. We also observed this trend in our RHA analyses which compared the claims-based LD

**Table 2. Incidence of Medically-Attended Lyme Disease Compared to Notifiable Disease Surveillance.**

| Category | 2010 | 2011 | 2012 | 2013 | 2014 | 2015 | 2016 | 2017 | 2018 | 2019 | 2020 | 2021 |
|---|---|---|---|---|---|---|---|---|---|---|---|---|
| **Overall incidence** | | | | | | | | | | | | |
| Medically-attended | 8.4 | 8.2 | 12.5 | 16.0 | 16.3 | 18.2 | 19.8 | 23.3 | 26.1 | 28.5 | 24.0 | 19.4 |
| Surveillance [2,5,20] | 1.0 | 1.0 | 1.5 | 2.3 | 2.7 | 2.4 | 3.9 | 3.2 | 4.0 | 4.8 | 2.2 | 3.0 |
| **Incidence male sex** | | | | | | | | | | | | |
| Medically-attended | 7.7 | 9.1 | 13.7 | 18.1 | 18.7 | 19.8 | 20.3 | 24.0 | 27.9 | 32.0 | 25.3 | 20.0 |
| Surveillance [21–24] | 2.1 | 2.1 | 2.1 | 2.1 | 2.1 | 3.2 | 4.8 | 3.4 | 5.0 | † | † | † |
| **Incidence female sex** | | | | | | | | | | | | |
| Medically-attended | 9.1 | 7.4 | 11.3 | 13.9 | 13.9 | 16.6 | 19.4 | 22.5 | 24.2 | 25.0 | 22.6 | 18.7 |
| Surveillance [21–24] | 1.5 | 1.5 | 1.5 | 1.5 | 1.5 | 1.2 | 3.0 | 2.9 | 2.9 | † | † | † |

† – Not Reported. Shaded cells represent average LD incidence reported for period 2010–2014.

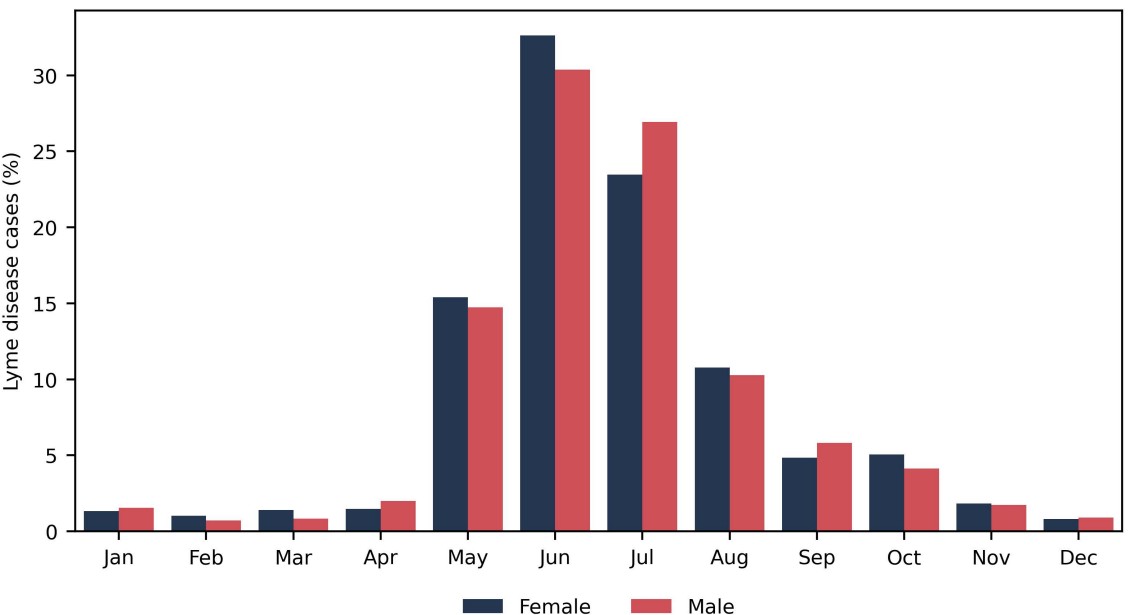

**Fig 1. Seasonality of Medically-Attended Lyme Disease.**

incidence to surveillance-reported LD cases, where RHAs with the lowest reported LD incidence had the highest under-reporting. Schwartz et al. (2023) also found higher under-reporting of claims-based LD cases among females, which was also observed in our study [11]. In Schwartz et al. (2023), female claims-based LD cases were more likely to be diagnosed outside of the Lyme season. This trend was not observed in our study (Fig 1).

The incidence of medically-attended LD in our study increased steadily from 2010 (8.4 per 100,000 population) to 2019 (28.5 per 100,000 population), followed by a decline in 2019−2021. This pattern was also observed with the surveillance-reported LD incidence in Manitoba [2,21–24]. During the same period, surveillance-reported LD cases increased in the neighbouring province of Ontario [25]. Provincial LD surveillance reports were unavailable in Manitoba from 2019−2021, nor were clinical LD reports reported to the LD enhanced surveillance program [4]. LD surveillance may have been impacted by the SARS-CoV-2 pandemic, by lack of access to primary care services for diagnoses and

reporting due to the diversion of public health resources. A US study found that the SARS-CoV-2 pandemic was associated with underreporting of LD cases, despite the increased utilization of green spaces while under lockdown [26]. Missed diagnoses and treatment in the early localized clinical stage during the COVID-19 lockdown, may lead to an increase in disseminated LD cases in 2022 [3].

Underreported LD incidence may have public health policy implications. The peak surveillance-reported LD incidence in Manitoba was 4.8 per 100,000 population in 2019 [2]. The claims-based LD incidence suggest that Manitoba LD incidence in each of the health regions except NRHA, have much higher incidence than reported.

An increase in LD incidence over time was observed in surveillance-based LD cases, Rusk et al (2025), and in the present study [13]. The increase may reflect the impact of climate change, environmental factors, and urbanization into forested areas, which favour tick habitat expansion, as well as increases in tick density in established areas, that increase opportunities for human-tick interaction [27]. A Canadian modeling analysis that evaluated the impact of various climate change scenarios on LD incidence, accounting for under-reporting of cases to public health, projected that the annual LD case count in Canada could increase to between 120,000 and 500,000 by 2050, compared to the approximately 3,000 cases currently reported annually [5,9].

There were several strengths to our study. The seasonal distribution of claims-based LD cases exhibited an expected pattern based upon peak tick activity, which has been linked to increases in human cases [27]. The inclusion of diagnostic serology results increased the overall claims-based LD case ascertainment by 15.8%. LD cases identified by this method may represent a higher risk of disseminated disease, since diagnostic and treatment codes were not identified at or before index, possibly indicating missed early diagnoses or patients who did not seek care. Sensitivity of diagnostic serology is low during the early localized stage and thus not generally recommended to diagnose LD at that stage [28].

Additionally, separating medically-attended LD cases by public health region allowed us to assess the distribution of LD across the province. Finally, the use of administrative data within a publicly funded healthcare system provides a comprehensive population-level assessment of LD diagnoses, with a large sample size that represents over 95% of the population. This reduces the risk of selection bias and exclusion of vulnerable populations, when compared to sample-based assessments within a healthcare system that has substantial private funding, as in the US.

There were several limitations of our study. The use of diagnostic codes and antibiotic dispensations could result in misclassification of LD. We were only able to identify medically-attended cases for which LD was investigated as a potential diagnosis which creates the risk of over-capturing cases that have been misdiagnosed, as well as missing cases that never underwent clinical investigation for LD. In addition, we were unable to assess the clinical stage of LD cases at index using administrative data, which limits our ability to assess the clinical impact of LD. It also limits our interpretation of the impact on vulnerable populations or to explain the impact of differential underreporting in women. We were unable to report LD cases by race, as we did not have demographic data on race. Difficulty diagnosing LD and the clinical impact of late diagnoses in patients with darker skin colour has been studied in the US. In both adults and children, African Americans were less likely to be diagnosed with EM, and more likely to be diagnosed with Lyme-related arthritis than white comparator groups [29,30]. Studies using other methods, such as electronic health records or active surveillance studies, should be conducted to evaluate the impact of race on diagnosis of LD in Canada. Prior to 2015, Manitoba used a 3-digit ICD-9-CA code for LD that also included other tick-borne diseases. To assess the validity of using the ICD-9-CA to identify LD cases before 2015, we conducted an assessment which determined that 95.6% of the cases previously identified by the 3-digit code, during the period April 1, 2015 – December 31, 2021, were LD (S11 Table in S2 File).

Underreporting cases of LD to public health surveillance was a substantial issue observed in our study. Causes of underreporting need further investigation but may be due to LD misdiagnoses, failure to seek health care, lack of reporting to public health surveillance due to systemic issues around administration of reporting or indifference/perceived lack of importance, or health care system failures, as may have occurred during the SARS-CoV-2 pandemic [31]. The clinical implications of missed or delayed LD diagnoses may cause substantial health burden to individuals and the healthcare

system [4,32]. Our study demonstrated that administrative data is an effective source to characterize the incidence of medically attended Lyme disease.

## Supporting information

**S1 File. Appendix A: S1-S10 Tables.**
(DOCX)

**S2 File. Appendix B: S11-S12 Tables.**
(XLSX)

**S3 File. Appendix C: List of Databases.**
(DOCX)

## Author contributions

**Conceptualization:** Maria Major, Kate Halsby, Frederick J. Angulo, Sarah J Willis.

**Data curation:** Maria Major, Susan Horton, Natalie Nightingale, Holly Yu, Irene Wang, Saranya Nair, Calum S. Neish, Doneal Thomas.

**Formal analysis:** Maria Major, Susan Horton, Natalie Nightingale, Kate Halsby, Doneal Thomas.

**Funding acquisition:** Maria Major, Frederick J. Angulo, Ana G Grajales.

**Investigation:** Maria Major, Sarah J Willis.

**Methodology:** Maria Major, Susan Horton, Natalie Nightingale, Kate Halsby, Frederick J. Angulo, James Stark, Holly Yu, Mark Loeb, Irene Wang, Saranya Nair, Calum S. Neish, Doneal Thomas, Sarah J Willis.

**Project administration:** Natalie Nightingale, Irene Wang.

**Resources:** Natalie Nightingale.

**Supervision:** Frederick J. Angulo, Ana G Grajales, Sarah J Willis.

**Validation:** Mark Loeb, Calum S. Neish.

**Visualization:** Chloe McDonald, Samuel Torres-Florez.

**Writing – original draft:** Maria Major.

**Writing – review & editing:** Susan Horton, Natalie Nightingale, Kate Halsby, Frederick J. Angulo, James Stark, Holly Yu, Mark Loeb, Irene Wang, Saranya Nair, Calum S. Neish, Doneal Thomas, Chloe McDonald, Ana G Grajales, Sarah J Willis.

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
