## [Decision Letter · Decision Letter 0]

1 Sep 2025

Dear Dr. Major,

Thank you for submitting your manuscript to PLOS ONE. After careful consideration, we feel that it has merit but does not fully meet PLOS ONE’s publication criteria as it currently stands. Therefore, we invite you to submit a revised version of the manuscript that addresses the points raised during the review process.

We look forward to receiving your revised manuscript.

Kind regards,

Stephen M. Rich, MS, PhD

Academic Editor

PLOS ONE

**Journal Requirements:**

1. When submitting your revision, we need you to address these additional requirements. Please ensure that your manuscript meets PLOS ONE's style requirements, including those for file naming. The PLOS ONE style templates can be found at https://journals.plos.org/plosone/s/file?id=wjVg/PLOSOne_formatting_sample_main_body.pdf and https://journals.plos.org/plosone/s/file?id=ba62/PLOSOne_formatting_sample_title_authors_affiliations.pdf 2. We note that the grant information you provided in the ‘Funding Information’ and ‘Financial Disclosure’ sections do not match.  When you resubmit, please ensure that you provide the correct grant numbers for the awards you received for your study in the ‘Funding Information’ section. 3. Thank you for stating the following financial disclosure: This study was sponsored by Pfizer.    Please state what role the funders took in the study.  If the funders had no role, please state: "The funders had no role in study design, data collection and analysis, decision to publish, or preparation of the manuscript." If this statement is not correct you must amend it as needed. Please include this amended Role of Funder statement in your cover letter; we will change the online submission form on your behalf. 4. Thank you for stating the following in the Competing Interests section: Maria Major, Sarah J. Willis, Kate Halsby, James Stark, Ana Gabriela Grajales, Holly Yu, Frederick J.  Angulo, Chloe McDonald are employees of Pfizer, and as part of their compensation, may hold Pfizer stock. Natalie Nightingale, Irene Wang, Calum Neish, Doneal Thomas, and Doneal Thomas are employees of IQVIA Solutions Canada Inc., which was a paid contractor to Pfizer in connection with the development of the study design and management. Saranya Nair was an employee of IQVIA Solutions Canada Inc. at the time of this study, which was a paid contractor to Pfizer in connection with the development of the study design and management. She is currently employed at Pfizer, Inc.Mark Loeb received compensation from Pfizer for his services as a member of the Lyme Disease Steering Committee.  We note that one or more of the authors are employed by a commercial company.  a. Please provide an amended Funding Statement declaring this commercial affiliation, as well as a statement regarding the Role of Funders in your study. If the funding organization did not play a role in the study design, data collection and analysis, decision to publish, or preparation of the manuscript and only provided financial support in the form of authors' salaries and/or research materials, please review your statements relating to the author contributions, and ensure you have specifically and accurately indicated the role(s) that these authors had in your study. You can update author roles in the Author Contributions section of the online submission form. Please also include the following statement within your amended Funding Statement. “The funder provided support in the form of salaries for authors, but did not have any additional role in the study design, data collection and analysis, decision to publish, or preparation of the manuscript. The specific roles of these authors are articulated in the ‘author contributions’ section.”If your commercial affiliation did play a role in your study, please state and explain this role within your updated Funding Statement.  b. Please also provide an updated Competing Interests Statement declaring this commercial affiliation along with any other relevant declarations relating to employment, consultancy, patents, products in development, or marketed products, etc.   Within your Competing Interests Statement, please confirm that this commercial affiliation does not alter your adherence to all PLOS ONE policies on sharing data and materials by including the following statement: "This does not alter our adherence to  PLOS ONE policies on sharing data and materials.” (as detailed online in our guide for authors http://journals.plos.org/plosone/s/competing-interests) . If this adherence statement is not accurate and  there are restrictions on sharing of data and/or materials, please state these. Please note that we cannot proceed with consideration of your article until this information has been declared. Please include both an updated Funding Statement and Competing Interests Statement in your cover letter. We will change the online submission form on your behalf. 5. When completing the data availability statement of the submission form, you indicated that you will make your data available on acceptance. We strongly recommend all authors decide on a data sharing plan before acceptance, as the process can be lengthy and hold up publication timelines. Please note that, though access restrictions are acceptable now, your entire data will need to be made freely accessible if your manuscript is accepted for publication. This policy applies to all data except where public deposition would breach compliance with the protocol approved by your research ethics board. If you are unable to adhere to our open data policy, please kindly revise your statement to explain your reasoning and we will seek the editor's input on an exemption. Please be assured that, once you have provided your new statement, the assessment of your exemption will not hold up the peer review process. 6. Please amend either the abstract on the online submission form (via Edit Submission) or the abstract in the manuscript so that they are identical. 7. Please include captions for your Supporting Information files at the end of your manuscript, and update any in-text citations to match accordingly. Please see our Supporting Information guidelines for more information: http://journals.plos.org/plosone/s/supporting-information. 8. If the reviewer comments include a recommendation to cite specific previously published works, please review and evaluate these publications to determine whether they are relevant and should be cited. There is no requirement to cite these works unless the editor has indicated otherwise. 

Reviewers' comments:

**Comments to the Author**

1. Is the manuscript technically sound, and do the data support the conclusions?

Reviewer #1: Partly

Reviewer #2: Yes

2. Has the statistical analysis been performed appropriately and rigorously?

Reviewer #1: N/A

Reviewer #2: Yes

3. Have the authors made all data underlying the findings in their manuscript fully available?

Reviewer #1: No

Reviewer #2: No

4. Is the manuscript presented in an intelligible fashion and written in standard English?

Reviewer #1: Yes

Reviewer #2: Yes

**Reviewer #1:** This is a very worthwhile study, but in my view it needs some clarifications before publication. This is a very worthwhile study, but in my view it needs some clarifications before publication. This is a very worthwhile study, but in my view it needs some clarifications before publication. This is a very worthwhile study, but in my view it needs some clarifications before publication.

Main point:

The main issue is that a number of different data sets are used to identify patients with Lyme disease diagnosis. These include patients diagnosed with Lyme disease and treated in ER, patients who were hospitalised, patients treated at primary care clinics and patients with a positive laboratory test result at the provincial laboratory. There is a potential for patients to be double or triple counted. For example, a patient who went to ER, had a positive test result, then deteriorated and had to be hospitalised for treatment, could result in a single case being counted as three cases. There is mention of comparing amongst these data for the earliest likely onset data, but there is no specific mention of unique identifiers for the patients that would mean that multiple records for the same patient do not result in overinflated case numbers. Please include description of this.

Minor points:

Mention in the text that test results are positive two-tier test results using CDC criteria for interpretation of western blots.

Much of Table 2 is not visible as it is too wide for portrait view.

There is no mention of boreal forests being unsuitable for ticks in ref 23. To my knowledge there have been no studies to explore if boreal forest is intrinsically inhospitable for I. scapularis. It is likely that boreal forest is not suitable due to absence or low densities of white tailed deer, which are essential hosts for adult I. scapularis.

The ‘gold standard’ estimation of under-reporting in the US is by Kugeler et al. 2021 (Emerging Infectious Diseases 27(2)) but there is no mention of this.

The section regarding CDC’s definition of an endemic area and discussion of low incidence and high incidence provinces needs to be removed. In Canada more detailed definitions of an endemic area are used and these are not based on incidence. There is no distinction of low- and high-incidence provinces in Canada as Lyme disease is an emerging infectious disease in Canada, with associated spatiotemporal variations in incidence.

**Reviewer #2:** This manuscript provides a valuable description of administrative health data, highlighting the increasing trend of Lyme disease cases over time in Manitoba, Canada. The paper is clearly written, and the analyses are rigorous. I have just a few minor comments. This manuscript provides a valuable description of administrative health data, highlighting the increasing trend of Lyme disease cases over time in Manitoba, Canada. The paper is clearly written, and the analyses are rigorous. I have just a few minor comments. This manuscript provides a valuable description of administrative health data, highlighting the increasing trend of Lyme disease cases over time in Manitoba, Canada. The paper is clearly written, and the analyses are rigorous. I have just a few minor comments. This manuscript provides a valuable description of administrative health data, highlighting the increasing trend of Lyme disease cases over time in Manitoba, Canada. The paper is clearly written, and the analyses are rigorous. I have just a few minor comments.

There appear to be inconsistencies regarding the province targeted in this project. The manuscript clearly relies on administrative health data from Manitoba; however, Ontario is mentioned in several instances.

• Abstract (lines 45–46): The text currently states, “Our study aims to estimate medically-attended LD incidence in Manitoba, Ontario, using administrative health data.” This should be corrected to “…Manitoba, Canada.”

• Supplemental document: The authors included a Research and Ethics Board (REB) Certificate from the University of Waterloo. However, the title of that certificate is “An observational study of Lyme disease in Ontario, Canada: Incidence and Healthcare Resource Utilization,” which does not correspond to the topic of the current manuscript.

It is understandable that the first author, as a University of Waterloo graduate, may have included this certificate as part of her PhD work. Nonetheless, attaching an REB certificate that references a different subject and geographic scope introduces confusion and does not add value to the present manuscript.

Lines 137–140, page 7: The manuscript states: “If individuals met more than one algorithm definition on the same day, they were indexed on the algorithm per the following order of precedence: emergency room or inpatient algorithm, primary care algorithm, serology-based algorithm.”

Could the authors clarify the rationale for establishing this specific order of precedence? In the same paragraph, they note: “If individuals met more than one algorithm definition during the selection period, they were indexed on the earliest day they met any of the algorithm definitions.”

Based on this rationale, one would expect the primary care algorithm to take precedence, as most Lyme disease patients initially consult their family physician during the early phase of illness. Typically, it is only when primary care fails to recognize the disease that it may progress to later stages, potentially leading to emergency room visits or hospitalizations.

Line 144–146, page 7: The age categories presented in the Methods section (5-year groups) differ from those in Table 1 (10-year groups). I recommend using consistent age group categories throughout the manuscript.

Line 167–169, page 8: The sentence “Per reporting rules for administrative health data in Manitoba, all outputs with a count of less than 6 patients were suppressed, along with the second smallest group to prevent back-calculation (i.e., double suppression)” duplicates the statement already included in the Data Privacy and Ethics Statement (lines 160–162). It does not need to be repeated in the Data Analysis section. Please remove it.

Line 172–173: The manuscript states: “Where possible, analyses were stratified by age group, RHA, and sex.” However, incidence is not presented by age group in any of the tables. I recommend removing “age group” from this sentence for consistency.

Table 2:

The value of presenting analyses by RHA is unclear, as data in more than two-thirds of the cells are suppressed or not reported. If the authors wish to retain the RHA-level analysis, I recommend combining the last two RHAs (Interlake-Eastern RHA and Northern RHA) to improve data presentation and interpretability.

In addition, the abbreviation DS used in the table is not defined in the footnote and should be clarified.

.

Reviewer #1: No

Reviewer #2: No

---

## [Author Response · Author response to Decision Letter 1]

15 Oct 2025

Please see revised manuscript and the letter entitled "response to reviewers" to read how each comment was addressed. I believe all comments were addressed in the revisions, but please let me know if you have any further questions or suggested revisions.

---

## [Decision Letter · Decision Letter 1]

7 Dec 2025

Dear Dr. Major,

Thank you for submitting your manuscript to PLOS ONE. After careful consideration, we feel that it has merit but does not fully meet PLOS ONE’s publication criteria as it currently stands. Therefore, we invite you to submit a revised version of the manuscript that addresses the points raised during the review process.

We look forward to receiving your revised manuscript.

Kind regards,

Jorge Cervantes

Academic Editor

PLOS One

**Journal Requirements:**

Reviewers' comments:

Reviewer's Responses to Questions

**Comments to the Author**

Reviewer #1: All comments have been addressed

Reviewer #2: All comments have been addressed

2. Is the manuscript technically sound, and do the data support the conclusions?

Reviewer #1: Partly

Reviewer #2: Yes

3. Has the statistical analysis been performed appropriately and rigorously?

Reviewer #1: N/A

Reviewer #2: Yes

4. Have the authors made all data underlying the findings in their manuscript fully available?

Reviewer #1: Yes

Reviewer #2: Yes

5. Is the manuscript presented in an intelligible fashion and written in standard English?

Reviewer #1: Yes

Reviewer #2: Yes

**Reviewer #1:**This manuscript is close to being ready for publication. My concerns regarding possible double counting of cases seems to have been mostly answered. However, some concerns about the inclusion of laboratory test data remain. Why would patients with positive test results not appear in the other databases, which identify clinical management? I cannot know – for this review by either the provincial laboratory director or a medical officer of Manitoba Health, Seniors & Long-term Care.This manuscript is close to being ready for publication. My concerns regarding possible double counting of cases seems to have been mostly answered. However, some concerns about the inclusion of laboratory test data remain. Why would patients with positive test results not appear in the other databases, which identify clinical management? I cannot know – for this review by either the provincial laboratory director or a medical officer of Manitoba Health, Seniors & Long-term Care.This manuscript is close to being ready for publication. My concerns regarding possible double counting of cases seems to have been mostly answered. However, some concerns about the inclusion of laboratory test data remain. Why would patients with positive test results not appear in the other databases, which identify clinical management? I cannot know – for this review by either the provincial laboratory director or a medical officer of Manitoba Health, Seniors & Long-term Care.This manuscript is close to being ready for publication. My concerns regarding possible double counting of cases seems to have been mostly answered. However, some concerns about the inclusion of laboratory test data remain. Why would patients with positive test results not appear in the other databases, which identify clinical management? I cannot know – for this review by either the provincial laboratory director or a medical officer of Manitoba Health, Seniors & Long-term Care.

Minor points:

In the introduction I would recommend using Borrelia burgdorferi sensu stricto rather than sensu lato. Early localised LD can appear as a rash, which can take the form of a bulls eye. I would maybe change ‘influenza type symptoms’ to something like non-specific malaise as there are no respiratory manifestations of Lyme disease.

In the methods it should be stated that the Discharge Abstract Database is a product of the Canadian Institute of Health Information (CIHI).

**Reviewer #2:**The authors have adequately addressed my comments raised in the first round of revision. I have no more comments.The authors have adequately addressed my comments raised in the first round of revision. I have no more comments.The authors have adequately addressed my comments raised in the first round of revision. I have no more comments.The authors have adequately addressed my comments raised in the first round of revision. I have no more comments.

.

Reviewer #1: No

Reviewer #2: No

---

## [Author Response · Author response to Decision Letter 2]

15 Jan 2026

Please see attached letter "response to reviewers" for detailed description of revision of manuscript in response to comments. Line numbers are provided for reference and refer to the unmarked manuscript line numbers.

---

## [Editor Report · Decision Letter 2]

20 Jan 2026

Use of Administrative Data for Evaluating Trends in Medically-Attended Lyme disease, Manitoba, Canada, 2010-2021

PONE-D-25-22049R2

Dear Dr. Major,

We’re pleased to inform you that your manuscript has been judged scientifically suitable for publication and will be formally accepted for publication once it meets all outstanding technical requirements.

Kind regards,

Jorge Cervantes

Academic Editor

PLOS One

Additional Editor Comments (optional):

Some pending minor issues pointed out by one of the reviewers need to be addressed.
---

## [Editor Report · Acceptance letter]

PONE-D-25-22049R2

PLOS One

Dear Dr. Major,

I'm pleased to inform you that your manuscript has been deemed suitable for publication in PLOS One. Congratulations! Your manuscript is now being handed over to our production team.

Kind regards,

on behalf of

Dr. Jorge Cervantes

Academic Editor

PLOS One